# Japanese Encephalitis Virus: An Update on the Potential Antivirals and Vaccines

**DOI:** 10.3390/vaccines11040742

**Published:** 2023-03-27

**Authors:** Kumar Saurabh Srivastava, Vandana Jeswani, Nabanita Pal, Babita Bohra, Vaishali Vishwakarma, Atharva Ashish Bapat, Yamini Prashanti Patnaik, Navin Khanna, Rahul Shukla

**Affiliations:** 1Division of Virus Research and Therapeutics, CSIR-Central Drug Research Institute, Lucknow 226031, India; 2Translational Health, Molecular Medicine Division, International Centre for Genetic Engineering & Biotechnology, New Delhi 110067, India

**Keywords:** Japanese encephalitis virus (JEV), Japanese encephalitis (JE), antiviral, vaccine, drug

## Abstract

Japanese encephalitis virus (JEV) is the causal agent behind Japanese encephalitis (JE), a potentially severe brain infection that spreads through mosquito bites. JE is predominant over the Asia-Pacific Region and has the potential to spread globally with a higher rate of morbidity and mortality. Efforts have been made to identify and select various target molecules essential in JEV’s progression, but until now, no licensed anti-JEV drug has been available. From a prophylactic point of view, a few licensed JE vaccines are available, but various factors, viz., the high cost and different side effects imposed by them, has narrowed their global use. With an average occurrence of >67,000 cases of JE annually, there is an urgent need to find a suitable antiviral drug to treat patients at the acute phase, as presently only supportive care is available to mitigate infection. This systematic review highlights the current status of efforts put in to develop antivirals against JE and the available vaccines, along with their effectiveness. It also summarizes epidemiology, structure, pathogenesis, and potential drug targets that can be explored to develop a new range of anti-JEV drugs to combat JEV infection globally.

## 1. Introduction

Japanese encephalitis virus (JEV) is the predominant cause of viral encephalitis in Asia [1]. It is a mosquito-borne flavivirus that is the causative agent of Japanese encephalitis [2]. Japanese encephalitis (JE) was reported for the first time in 1871, in Japan, and JEV was first isolated in 1935 from the brain of a fatal case of JE. This isolate, known as the Nakayama strain, is acknowledged as the prototype strain of JEV [3]. Other clinically relevant viruses belonging to the same genus include dengue virus (DENV), yellow fever virus (YFV) Murray Valley encephalitis (MVE), West Nile virus (WNV), zika virus (ZIKV), St. Louis encephalitis virus (SLEV) and tick-borne encephalitis virus (TBEV) [4]. According to a WHO report published in 2019, almost 68,000 cases of JE with a 20–30% mortality rate were recorded annually. A study based on mathematical modeling with age-stratified case data estimated that approximately 100,308 clinical cases and 20,000–30,000 deaths occurred due to JEV in 2015 across the globe [5]. Newborns and children up to the age of 15 years are more vulnerable to JE with the increased threat of neurological complications over adults [6]. Almost 2 billion people living in endemic countries face a constant threat of JE and an upsurge in the mosquito population poses a risk of expansion to newer geographical areas.

The JEV is an RNA virus, primarily transmitted through the bite of an infected female mosquito, *Culex tritaeniorhynchus*. Other *Culex* species, such as *Cx. annulirostris*, *Cx. vishnui*, *Cx. pseudovishnui*, *Cx. gelidus*, *Cx. sitiens* and *Cx. fuscocephela* are also reported to be involved in the transmission of JEV along with some *Anopheles* mosquito species, such as *Anopheles subpictus*, *An. peditaeniatus* and *An. hyrcanus* [7]. The primary reservoirs of JEV are the birds of the family *Ardeidae*, such as herons and egrets. Pigs are highly susceptible to JEV, where the virus becomes amplified in optimum levels and they develop a high circulating viral titer (amplified host), and are therefore able to spread the infection to naive mosquitoes [8]. Reports suggest that at this stage, pigs tend to shed the virus in oronasal secretion and may potentiate the horizontal transmission of JEV [9]. Unlike pigs and birds; humans, cattle and horses do not develop high viral titers, making them ‘dead-end’ hosts, as shown in Figure 1.

Nonetheless, JEV has an enzootic cycle, due to which the virus can persist in nature to such an extent that it might be next to impossible to eradicate it in the near future. Thus, effective antiviral therapy and an ideal vaccine against JEV are of pressing priority. Despite challenges, such as the lack of an appropriate drug delivery system or a treatment plan independent of the stage of infection, JE research has increased rapidly with growing technological advancements, with the primary aim of developing a safe and cost-effective therapeutic (drugs and vaccines) for all age groups. This review thoroughly discusses the recent trends in the discovery of JEV-targeted compounds that have the potential to be developed as therapeutic drugs and the efforts that have been made toward the development of effective vaccines against JEV.

## 2. Epidemiology

The majority of cases of viral encephalitis in the Asian subcontinent are due to JEV [6]. This spans a large region that includes majorly tropical parts of Asia, such as Japan, China, Taiwan, Korea, the Philippines, India and all of Southeastern Asia. Countries with confirmed JE epidemics include India, Nepal, Pakistan, Sri Lanka, Myanmar, Laos, Vietnam, Malaysia, the Philippines, Singapore, China, Indonesia, maritime Siberia, Japan and Korea [10]. Additionally, sporadic outbreaks in the Western Pacific and northern Australia are also observed [6]. Historically, outbreaks similar to JE had been recorded in Japan in the late 1800s; however, the first confirmed JE case was documented in 1924 in Japan, followed by Korea (1933), China (1940), the Philippines (1950), India (1955) and many other Asian countries thereafter [5]. In recent decades, geographical hotspots for JE incidences have shifted considerably from South Asian countries (e.g., Japan, South Korea and Taiwan) to South East Asian countries, including Bangladesh, Cambodia, India, Indonesia and Pakistan (Figure 2).

Since the early 1970s, there has been a rise in epidemic activity of JEV in the Indian subcontinent. Then, in late 1990s, the virus persisted to spread into the neighboring territories of southern Pakistan along with the Kathmandu Valley of Nepal [10].

In India, the first evidence of JE was obtained through serological studies back in 1952, and the first JEV case in India was documented in 1955. Later on, frequent outbreaks were reported in all parts of India at regular intervals [10,11]. A major outbreak occurred in the Bankura district of West Bengal in 1973, with a fatality rate of 42.6% [10]. The most prolonged epidemic of JE was witnessed in the Gorakhpur district of Uttar Pradesh in 2005, with more than 5500 documented cases of viral encephalitis and ~23% fatalities [10,12,13]. Several reports indicated the expansion of the virus to the newer non-endemic areas, including the northern and northeastern parts of the Indian continent, and cases have been reported that signify the spread of the virus, including in urban areas, such as New Delhi [14,15].

More recently, JEV infection has made a threat of resurgence that was highlighted by the fairly large epizootic outbreak in Australia in 2022 [16]. In Australia, the first JE case was identified in 1995 and the virus has remained dormant over the past two decades [17]. In early 2021, it reappeared and was diagnosed in the resident of the northern territory in Queensland, which resulted in death, and there appeared to be a sentinel event in the recent outbreak in 2022 in the southern Australian states. Then, JEV was detected in stillbirths, mummified fetuses and newborn piglets from several commercial piggeries majorly located in the four southern states of Australia (New South Wales, Queensland, South Australia, and Victoria), Figure 2. As per the Australian government, 45 human cases of JEV have been notified up to 5 January 2023, of which 35 are confirmed cases with conclusive clinical evidence and seven fatalities [16,18,19]. The primary causative agents of the JE transmission were the members of the subgroup *Culex sitiens*, particularly *Cx. annulirostris* [9]. The other *Culex* species, *Cx. quinquefasciatus*, *Cx. gelidus* and *Cx. tritaeniorhynchus* were suspected to transmit the disease in Australia [18], although more research and mosquito surveillance needs to be carried out to confirm their geographic distribution and abundance in Australia. Nonetheless, current JEV infections in the southern Australian states pose a significant risk to their neighboring territorial states, such as the Northern Territory and Western Australia.

## 3. JEV Structure and Its Genome

Japanese encephalitis virus is an enveloped, positive-sense single-stranded RNA virus measuring ~40–50 nm in diameter with spheroid cubical symmetry. The viral RNA genome (~11 kb) encodes genes for three structural (C: capsid; prM: precursor membrane and E: envelope) and seven non-structural (NS) proteins (NS1, NS2A, NS2B, NS3, NS4A, NS4B and NS5) with the 5′methylated cap, but is devoid of a poly-A tail at 3′ end. All ten of these proteins encoded from a 3432 amino acid long chain are translated from a single open reading frame of genomic RNA.

The capsid (C) structural protein of JEV dimerizes head-to-tail in an anti-parallel manner. Multiple copies of capsid dimers tend to compose in a spherical nucleocapsid, enclosing the viral genomic RNA. Recently, the crystal structure of the C protein was revealed and it was shown to have α-helixes 1–4 secondary structure, which closely resembles the capsid protein of DENV, WNV and ZIKV [20]. Each monomer of the JEV capsid protein consists of four helices: α1 (amino acid 29–38), α2 (amino acid 44–57), α3 (amino acid 63–70) and the longest α4 (amino acid 74–96), connected by short loops. The amino-terminal of α helix-1 forms a closed and open confirmation by which it tends to be flexible, and this makes it an attractive antiviral drug target. However, the carboxyl-terminal pairing of α helix 4-4 arrangement leads to a coiled-coil-like structure, which possibly helps in nucleic acid binding.

The viral precursor membrane or pre-membrane (prM) emerges from nascent polyprotein following co-translational cleavage by a signal peptide and starts to assemble at the viral genomic RNA containing vesicles of the ER. Soon after it is assembled, the vesicle buds from the ER and reaches the Golgi body network, where prM is cleaved by furin enzyme into the M protein and forms a mature virus particle before its release from the host cell. The exact function of prM/M in JEV is still unknown. However, in other related viruses of the same family, including DENV and YFV, it is well studied, where it helps in the assembly of mature viruses and provides a moiety for the proper conformational arrangement of the E protein. Furthermore, the membrane protein of the virus plays a crucial role in the binding of the E protein domain with the host receptor while establishing the infection. Following the virion internalization, the membrane protein assists in the fusion of viral vesicles to the endosome.

The envelope protein (E) of JEV is an essential protein that covers the virion and is responsible for binding to the host cell receptors and fusion with the host plasma membrane. The E protein of JEV is the major protein recognized by virus-neutralizing antibodies. Similar to the E protein of other flaviviruses, it is composed of three domains, envelope domain I (EDI), envelope domain II (EDII) and envelope domain III (EDIII). The crystal structure of the E protein has revealed that it dimerizes antiparallelly in a head-to-tail manner, as observed in other flaviviruses, but with a relatively small interface [21]. The EDI forms the central domain of the envelope with a nine-stranded β-barrel which is located between the extended EDII and globular domain EDIII. EDII is basically formed with two extended loops that extend from EDI and are stabilized by binding with three disulfide bonds, conserving its fusion peptide at the top. Similar toother flaviviruses, EDIII retains immunoglobulin-like structures at the carboxyl terminus of the ectodomain which tends to bind to the host receptors for its virion internalization. These properties of the EDIII domain make it a potential target for the designing of antivirals for mitigation of JEV infections.

All seven non-structural (NS) proteins are translated and cleaved from the single polyprotein and are majorly involved in the replication, and assist in the assembly of new viral particles. The NS1 of JEV is relatively not well studied and its exact function is still unknown. Some studies indicate that NS1 is involved in the replication complex for the transcription of the double-stranded intermediate RNA [22]. A few studies have demonstrated that small interfering RNA (siRNA) against NS1 could inhibit the replication of JEV in vitro [23]. However, none of the siRNA were carried forward for clinical development. Furthermore, a study by Yen et al., has indicated that NS1 acts as a heterologous epitope and induces a cross-reactive immune response against various pathogens [24]. The NS2A protein has direct involvement in the viral RNA genome synthesis and its assembly and NS2B forms the complex with NS3 and helps in the viral protease activity. NS3, however, acts as helicase and aids in the genome replication. It is also believed that NS3 has originated from the ER or trans-Golgi network and works as a reservoir for viral proteins during virion assembly [25]. The NS4 protein is highly hydrophobic and is supposed to participate in constructing the membrane component. Due to this fact, it is believed that it might play a crucial role in the adaptability of the virus to different environments, though the exact function of both NS4A and NS4B is still unclear. The NS5 is a multi-enzymatic protein and a vital component of the viral RNA replication complex. Similar to nonstructural and cellular proteins of other flaviviruses, NS5 carries both methyl transferase domain in the N-terminus and the RNA-dependent RNA polymerase (RdRp) domain in the C-terminus [26]. The methyltransferase domain of NS5 is largely responsible for the 5′ capping of viral genomic RNA and the RdRp domain directly plays a role in RNA replication [27,28,29,30].

## 4. JEV Genetic Diversity

Sequencing of complete and partial viral genomes has led to the identification of prevalent viral genotypes. Five different JEV genotypes have been identified (GI-GV) so far. The genotypic distribution of JEVs has an observable geographic pattern, with (a) the Indonesia–Malaysia region having all five isolated genotypes; (b) GI and GII are found in the Australia–New Guinea region; (c) GII and GIII are prevalent in the Taiwan–Philippines region; (d) the Thailand–Cambodia–Vietnam region have the GI, GII and GIII genotypes; (e) the Japan–Korea–China region have GI and GIII; and (f) the India–Sri Lanka–Nepal region having GIII. However, several reports suggested that GI is replacing GIII as the dominant genotype in India. Detailed phylogenetic analysis of the present strains suggests the origin of JEV came from an ancestor in the Indonesia–Malaysia region and diverged into five different genotypes. Genotypes GIV and GV, which still circulate in the region are older forms and the newer ones, which are more recent in evolutionary history, viz., GI, GII and GIII, have spread to other parts of the world. A group of GV has been isolated in China and South Korea post-2008, which indicates that they emerged recently and circulated outside the Indonesia–Malaysia region. Although all JEV genotypes form a single serotype, at least five antigenic groups are differentiated by various immunological assays, demonstrating some degree of antigenic variation among circulating JEVs [31]. Thus, the genetic and antigenic heterogeneity of the JEV may have a significant impact on JE prevention and control. Genotypes of JEV, along with the most affected countries, are given in Figure 3 [32].

## 5. JEV Pathogenesis

The primary requirement for pathogenesis is that viremia in the reservoir host must be high enough for a biting naive mosquito to acquire the virus [6]. Once a mosquito sucks blood from an infected host, the virus has to be passed from various physical and physiological barriers before it is replicated [33]. Once the virus reaches the midgut, the viral envelope fuses with the plasma membrane of intestinal epithelial cells and releases its genome into the cytoplasm, if the peritrophic membrane (a potential physical barrier) is not formed yet. Thereafter, the virus is replicated and virions burst out into the hemocoel, from where it travels to the tracheal system and finally reaches the salivary glands [33]. Eventually, the acinar cells of the salivary glands become infected with the virus after passing the salivary gland infection barrier and begin to shed virions into the saliva and the carrier mosquito is ready to infect the subsequent host when it bites. When an infected mosquito bites a healthy human, the virions are released into the dermal cells, from where the virus causes either latent infection by infecting only mononuclear cells, or persistent infection in which it invades the nervous system [34,35]. Pathogen-associated molecular patterns (PAMP) bind to pathogen recognition receptors (PRR) to induce interferon stimulatory genes (ISGs, such as PKR, OAS, TRIM21, ISG15 and MX1) via the JNK pathway. These interferons induce the antiviral state in the infected host cell and neighboring uninfected host cells, but the virus can change it in its favor. The virus primarily invades innate immune cells in the skin, such as keratinocytes, Langerhans cells, fibroblasts, dermal dendritic cells and endothelial cells [36]. The dendritic cells are the primary target for JEV infection and potentiate the secretion of pro-inflammatory cytokines, such as IL6, IL8, IL12 and TNF-α, and attract other immune cells by which the virus has the opportunity to enter the draining lymph nodes and secondary lymphoid organs, including the spleen, followed by other peripheral tissues, such as the kidney, liver, heart, and lungs [37]. The virus replicates in tissue macrophages and monocytes in the blood [38]. If the immune system generates a humoral response (IgM) within the first 5 days of the virus incubation period (sub-clinical stage) then virus may be cleared before reaching the central nervous system [36]. If not, the most dangerous clinical phase commences as the virus then begins to enter the central nervous system after breaching the blood–brain barrier (BBB). There are various ways of breaching the BBB, (a) direct diffusion through endothelial cells, infecting them by replication; (b) Trojan horse mechanism, in which the virus enters cerebrospinal fluid (CSF) by colonizing in the inflammatory cells or blood lymphocytes; (c) chymase release by mast cells or metalloproteases (MMP 2/MMP 9) which loosens the tight junction between endothelial cells [39]; or (d) receptor-mediated endocytosis [35,36]. All of these mechanisms can kill neuronal cells directly or by causing neuronal bystander death. The entry of the virus in the CSF activates microglial cells, astrocytes and pericytes due to the excessive release of pro-inflammatory cytokines, such as COX-2, iNOS, IL-6 and TNF-α, which causes inflammation in some parts of the brain (neuroencephalitis) [30]. Studies report that IFN-α (anti-inflammatory cytokine) provides immunity against JEV by inhibiting replication at various stages, as well as assembly and release. Its efficacy has been studied in vitro but a similar effect was not observed in humans [40,41].

Furthermore, JEV enters neuronal cells using PLVAP and GKN3 receptors [42]. Subsequently, it lowers the levels of anti-inflammatory cytokines (IL-10 and IL-4) and also kills neuronal cells [35]. Carnage of the neurons stimulates astrocytes and microglial cells to release inflammatory mediators which cause further neuronal death [43]. Astrocytes and microglial cells also release chemokines (IP-10, RANTES, IL-8 and MCP-1) which facilitate the entry of blood lymphocytes in the CSF, further elevating the infection [44]. The proliferation of neuron progenitor stem cells (NPSC), that are responsible for the immune response into CSF, is inhibited due to virus infection. The prolonged inhibition of protein synthesis due to interferons may be lethal to healthy neuronal cells. Studies have demonstrated that individuals with a higher concentration of anti-JEV IgM in CSF have more chances of survival without much effect on the CNS [44,45].

Interaction of the virus with host receptors includes recognition, attachment, binding and entry. The process initiates with the association of the virus to attachment factors, such as heparansulfate proteoglycans, which brings it to the vicinity of the target host cell until it finds a suitable receptor on the plasma membrane. The virus interacts with several host receptors, such as heat shock protein 70, CD4, α5β3 integrin, dendritic cell-specific intercellular adhesion molecule-3-grabbing non-integrin (DC-SIGN), glucose-regulated protein 78 (GRP78), T-cell immunoglobulin and mucin domain 1 (TIM-1), etc. [36]. Following the establishment of the attachment with the host receptor, the viral particles are internalized by clathrin-dependent and independent endocytosis in fibroblasts, epithelial cells and neuronal cells. Conformational changes in the E glycoproteins occur in an endocytic vesicle, which releases hydropathic patches, resulting in the fusion of the envelope with an endosomal membrane (Figure 4). Thereafter, the virus uncoats and releases its single-stranded positive-sense RNA genome in the cytoplasm under the influence of acidic pH [46]. Non-structural proteins of the virus along with host factors and endoplasmic reticulum-associated protein degradation (ERAD) proteins, such as LC3-1, EDEM 1 and SEI1L interact to form a replication complex at the surface of the endoplasmic reticulum [35,47,48]. ssRNA replicates into a double-stranded replicative form, which serves as a template for the synthesis of positive sense mRNA. This is followed with viral protein synthesis and processing by host and viral proteases. Virion assembles in the ER lumen and matures via the cleavage of prM to M by the host furin at trans-Golgi followed by its release [36] (Figure 4).

## 6. Clinical Manifestations

In general, most JE infections are mild with subclinical febrile illness or without any evident symptoms. However, among the patients who develop severe clinical illness (encephalitis), the case fatality rate can be as high as 30%. About 30–50% of the patients who survive severe infection face behavioral or neurological sequelae, such as paralysis, frequent seizures and/or inability to speak [10,49]. The progression of JE is divided into three stages in patients who progress to neurological disease. The first stage, the prodromal phase, is marked by mild fever, chills, muscle pain, meningitis, vomiting and diarrhea. In the second stage, the acute phase, a person suffers from reduced consciousness, seizures, parkinsonian syndrome which further manifests into viral encephalitis in which a person may experience tremors, generalized hypertonia, cogwheel rigidity and other abnormalities that may even result in a coma. In some cases, the disease progresses rapidly and may be lethal. In the third stage, the late phase, the person either recovers or suffers from elongated neurological sequelae [35,43]. Several parts of the brain, such as the hippocampus, thalamus, basal ganglia, parenchyma, cerebral cortex, midbrain, brain stem, temporal lobes, substantia nigra and anterior horn cells of the spinal cord become affected among which the hippocampus is found to be majorly affected. Few histopathological changes observed are perivascular cuffing, vascular leakage, microglial nodule formation, glio mesenchymal nodules, necrolytic lesions, scarred ramified foci, cerebral edema and congested leptomeninges [35].

## 7. Potential Drug Targets

Various proteins, crucial for viral attachment, replication and maturation, can be considered as potential drug targets. Data suggest that developing host-related proteins as drug targets can be more advantageous than targeting the viral proteins, as it allows for targeting multiple viruses simultaneously [50]. It reduces the chances of drug resistance and maintains drug effectivity, even if the virus is mutated. However, targeting host molecules might be disadvantageous as it may induce various levels of cytotoxicity and side effects at the cellular level [51,52]. Analysis of the JEV lifecycle has also identified various stages at which the replication of the virus can be inhibited via targeting different structural and non-structural proteins. Structural proteins, including capsid and envelope proteins, and Non-structural proteins, such as NS3 and NS5, can be explored as unique drug targets [30].

The C protein dimerizes and its C-terminus associates with the viral RNA to form a nucleocapsid (NC) [53,54]. Such an association allows for the stabilization of viral RNA; thus, molecules inhibiting the dimerization of the C protein or the RNA-protein interaction can be developed as anti-JEV drugs [20]. The prM and E proteins which are the main constituents of the immature virion, inhibit the premature budding of the virus particles [55]. The maturation of the virions is a result of the conformational changes of the major surface component dimeric E-protein by the cellular serine protease furin [55,56]. Studies indicate that the N linked site of domain I of the E protein interacts with cellular receptors and can be concluded for the infectivity of the virus [21,57]. Structural analysis has suggested various sites of the target in the E protein, such as the β-OG ligand binding pocket in the fusion loop of EDII, the E-protein rafts in mature virus and the E homotrimers in post-fusion state [55,56,57].

The proteolytic cleavage of NS2A-NS2B, NS2B-NS3, NS3-NS4A and NS4B-NS5 are essential for the assembly of the viral replicase complex [56]. This is processed by a heterodimeric complex of NS2B-NS3. NS3 has a serine protease domain at the N-terminus, which is enzymatically inactive, and is active in association with NS2B. NS2B contributes to the essential folding of the NS3 protease site [58,59]. The C-terminus of the protease and helicase domain of NS3 has seven conserved motifs of NTPase and RNA helicase [58]. NS3 helicase performs vital viral replication functions, such as the DNA duplex resolution, aids in RNA synthesis initiation by removing the protein bound to the genome and the melting of secondary structures [60,61]. It also has the ATPase activity, which regulates the ATP-dependent strand separation. Similarly, all NS3 helicases have NTPase activity to hydrolyze nucleoside triphosphate non-specifically, to meet the energy requirement of the replication process [61]. The N-terminus of NS5 has methyltransferase activity which regulates the 5′ capping of nascent RNA by methylation of the 5′ guanine cap and the 2′hydroxyl group of ribose sugar. Mutation at this site is known to cause impairment of viral replication [62]. The C-terminus of NS5 has RNA-dependent RNA polymerase (RdRp) activity, which initiates the RNA synthesis in the absence of primers [63]. The absence of RdRp in humans makes RdRp a propitious drug target. Accordingly, viral proteins, such as NS2B-NS3 protease, NS3 helicase, NS5 methyltransferase and NS5 RdRp, warrant further research as potential drug targets for the development of anti-JEV drugs. Apart from that, various host factors, such as ornithine decarboxylase, histone deacetylases and HSP70 have been identified to play a crucial role in advancing viral infection. These several phenomena can be exploited to inhibit the infection of JEV and hence should be explored further.

## 8. Potential Antivirals against JEV

Japanese encephalitis is one of the most serious infections, endemic to Asia which is wreaking havoc and the irony is that no specific antivirals are available to date for its treatment. Current therapeutic strategies do not target attenuating the virus although, it is the only treatment for various clinical manifestations. Innumerable drugs have been investigated (Table 1), but none have been found to be effective to date, so the journey to the discovery of specific drug against JEV continues.

### 8.1. Broad Spectrum (Non-Specific) Antiviral Molecules Used against JEV

Scores of broad-spectrum antivirals have been studied to inhibit JE. Some of them were identified to inhibit the infection actively. Rosmarinic acid has been observed to reduce the viral replication of the GP78 strain of JEV in mice brains [64,65]. Curcumin, a phytochemical which is also an antioxidant, has been shown to reduce new virus formation via the dysregulation of the ubiquitin protease system [30,66]. Minocycline has demonstrated a reduction in virus titer, impediment of neuronal apoptosis and microglial activation in in vitro and in vivo studies [67]. It has also shown effects on the protection of the blood–brain barrier which usually becomes impaired during infection [68,69]. All of these drugs can be tested for their specific activity against the JEV. Interferons and interferon inducers, such as aloe-emodin, have also displayed inhibitory activity against JEV infection by triggering an adaptive immune response [70,71]. Ribavirin, an inhibitor of guanine nucleotide synthesis, which targets inosine monophosphate dehydrogenase, has been tested in children, but it exhibits an almost negligible effect in treating JEV [72,73,74,75]. HSP70, due to its upregulation and its function during JEV infection, its inhibitors, such as apoptozole, have also been identified to inhibit JEV in vitro [76] (Table 1).

### 8.2. Nucleic Acid Based Anti-JEV Molecules

Developments in micro-RNA-based technologies has provided an advantage to explore nucleic-acid based drugs designed against JEV. Targeting of the viral genome with miRNA inhibits the propagation of the virus at transcription and translation levels [77]. Although they lack specificity with different strains and require simultaneous administration, they have provided a promising outcome in inhibiting JEV infection in in vitro and in vivo studies [29,63,77]. Several studies have indicated an efficient reduction in JEV infection by targeting genes for various essential proteins by miRNA [78]. In addition to miRNA, shRNA can also be used to silence a part of the viral genome. Studies have demonstrated effective viral inhibition by targeting E, NS4b and C genes utilizing shRNA [79]. Other aspects of nucleic acid-based drugs are morpholino oligomers and peptide nucleic acids (PNAs). PNAs are peptide-like backbones containing nucleic acid derivatives with side chains of heterocyclic bases. Morpholino oligomers are DNA bases attached to methylene morpholine ring backbones linked to phosphorodiamidate groups. They both have the ability to irreversibly bind to complementary sequences with high specificity, hence they can be used to inhibit the viral replication binding at specific sequences of the viral genome [80,81,82] (Table 1). Further studies using nucleic acid inhibitors assure a promising therapeutic potential for JE.

### 8.3. Replication Cycle-Based Anti-JEV Molecules

Inhibition of the viral replication at different stages can be a potential target for a drug identification. Starting from the entry to maturation and release of viral progenies; for every step, inhibitors could be designed that may inhibit a particular step, resulting in the inhibition of viral replication. Different approaches have been made to inhibit the attachment and entry of JEV. Proteoglycans, including heparin sulfate and chondroitin sulfate, that serve as essential cellular receptors, have been identified as potential targets against JEV [29]. Their derivatives have also been reported to provide partial protection against JEV [83,84,85]. Bovine lactoferrin binds to the heparin sulfate receptors and prevents the attachment of the virus to the cells [86].

Viral RNA replication is governed by several factors and numerous compounds have been tested to target it and have provided effective results in eliminating the infection. MCPIP1 [monocyte chemoattractant protein 1-induced protein 1] has a nuclease domain that has been observed to express anti-JEV activity in vitro. It targets various RNA sites and inhibits replication [87,88]. A phytochemical, pokeweed protein extracted from *Phytolacca americana*, exhibits depurination of viral RNAs. Inhibition of JEV was observed in mice after treatment [89]. Kaempferol, a natural flavanol, has also exhibited inhibitory activity against JEV by neutralizing the virus via binding to a frameshift site of viral RNA [90] (Table 1).

Advancements in structural virology and in silico methodologies have eased methods to identify new inhibitors against viral targets, primarily against NS3, NS5 and E proteins due to their essentiality in viral replication and infection. In an in-silico study, bortezomib was identified to target the genome of JEV [91]. Various compounds/drugs have been studied through computational approaches of drug discovery and analyzed via advanced high throughput technologies. Although they had varied efficacy levels in in vitro and in vivo studies, many of them failed to exhibit a similar effect in humans or they were inefficacious in clinical studies [30]. Hence, the journey to find some promising drug against JEV continues.

**Table 1 vaccines-11-00742-t001:** Antivirals being explored against JEV.

Type of Drug/Drug Target	Compound/Drug Name	Mechanism of Action	In Vitro Activity: IC_50_ or EC_50_ *(Utilised Cell Line)*	In Vivo *Efficacy:* *% Survival (Explored Animal Model)*	References
**Broad use**	Arctigenin	Anti-oxidative/anti-inflammatory effect	3.9µM (U937 Cells)	100% (BALB/c)	[92,93]
**(anti-inflammatory)**	Fenofibrate		ND *	80% (BALB/c)	[94]
	Rosmarinic acid		ND	80% (BALB/c)	[64]
	Diethyldithiocarbamate (DDTC)		ND	100% (swiss albino)	[95]
	Astragali radix extracts		ND *	>80% (ICR)	[96]
	Lacidipine		3.5 µM (Vero cells)	ND	[28]
	Tilapia hepcidin 1–5		1 µg/mL (BHK-21 cells)	DM ^#^ (C3H/H3N)	[97]
	Curcumin		ND *	DM ^#^ (BALB/c)	[30]
**Immune system based**	Aloe-emodin	Triggers adaptive immune responses to generate an antiviral state	0.50 µg/mL–1.51 µg/mL (BHK 21 cells)	ND	[71]
	Interferons		ND * (LLC-MK2 cells)	ND^	[98]
	Enanderinanin J	Inhibits autophagosome-lysosome fusion	16.3 µM (A549 cells)	ND	[97]
	Atorvastatin	Reduces secretion of pro-inflammatory cytokines by the neurons and causes neuronal death by evading the miR-21 upregulation, which is induced by the virus in a hn-RNPC-dependent fashion	ND *	ND	[99]
	Pimecrolimus	Blocks T-cell activation	3.1 µM (Vero cells)	ND	[28]
	Bafilomycin A1	Inhibits pH-triggered membrane fusion of the endocytosed JEV and vacuolar type proton pump	ND *	ND	[86]
	Artemisinin	Enhances the host type I interferon response	18.5 µM (A549 cells)	~50% (C57BL6J)	[100]
**Cell signaling based**	Aspirin, indomethacin, sodium salicylase	Inhibits cyclooxygenase, modulates the intracellular MAP kinase pathway followed by JEV infection	ND *	ND	[65]
	Dehydroepiandrosterone (DHEA)	Upregulates MAPK pathways; induces ERK activation	ND *	ND	[101]
	AR-12	Inhibits PI3/AKT pathway and GRP78; inhibits mitochondrial enzyme DHODH (dihydroorotate dehydrogenase)	~509.9 nM (A549 cells)	ND	[102,103]
	P12-23 (Derivative of AR-12)		~53.2 nM (A549 cells)	ND	
	P12-34 (Derivative of AR-12)		~56.1 nM (A549 cells)	ND	
	Anisomycin	Restores function of ERK (extracellular signal-regulated kinase); suppresses JEV induced cytotoxicity	ND *	ND	[65,101]
**Host factors targeting**	Eflorinithine	Inhibits polyamine biosynthesis	ND *	ND	[104]
	Tubacin	Inhibits histone deacetylases	1.52 µM (TE671 cells)	ND	[105]
	Mitotane	Deregulates cytochrome P450 enzymes	6.6 µM (Vero cells)	ND	[28]
	Digoxin and ouabain	Targets the Na^+^/K^+^-ATPase	<0.1031 µM (Vero cells)	~60% (BALB/c)	[106]
	Benidipine hydrochloride	Inhibits the triple calcium channel (L, N, T type calcium channels)	3.7 µM (Vero cells)	ND	[28]
	Berbamine	Blocks TRPMLs to compromise endosomal trafficking of LDLR, decreases its level of plasma membrane, thus blocking JEV entry	1.62 µM (A549 cells)	~75% (BALB/c)	[107]
	BCX4430(galidesivir)	Inhibits replication	43.6 µM (Hela cells)	ND	[108]
	Apotozole	Inhibits HSP70	ND *	ND	[76]
**Nucleic acid analogues**	miRNA	Binds and inhibits genes coding for proteins such as, the E protein Domain II, NS5, capsid (C), membrane (M) protein, envelope(E), prM, NS1, NS2A, NS2B, NS3, NS4A and NS4B	ND *	DM ^#^ (BALB/c)	[79,109,110]
	shRNA	Binds and inhibits genes of E, C and NS4B proteins	ND *	50–70% (BALB/c)	[79]
	PNA (J3U5)	Targets the 5′ untranslated region of JEV genome	ND *	ND	[80]
	Morpholino oligomers (PPMO-P10882)		ND *	60–70% (BALB/c)	[81]
	DNAzymes	3′ Non-coding sequence of JEV genome	ND *	100% (BALB/c)	[111]
**Viral entry and attachment Inhibitors**	Bovine lactoferrin	Binds to heparin sulfate receptors; prevents attachment	26.1 µg/mL; 518.3 µg/mL (BHK21 cells) ^$^	ND	[112]
	Griffithsin	Binds to the E protein; prevents attachment	265 ng/mL (BHK21 cells)	ND	[89]
	Curcumin carbon quantum dots	Binds to the E protein, prevents viral entry into the host cells	0.9 µg/mL; 100 µg/mL (BHK21 cells) ^$^	ND	[113]
	Carrageenan (sulfated polysaccharide)	Inhibits entry into host cells	15 µg/mL (WRL68 cells)	ND	[114]
	E-protein domain III binding peptide	Inhibits the E-protein and receptor interaction	1 µM (BHK21 cells)	ND	[115]
	Monoclonal antibodies (2F2; 2H4)	Blocks the virus-receptor attachment	~1.4 ng/mL (Vero cells)	100% (BALB/c mice)	[116]
	Indirubin	Inhibits the viral attachment	11.79 µg/mL >50 µg/mL (BHK21 cells) ^$^	>50% (BALB/c mice)	[117]
	Indigo				
	Heparin		10µg/mL (BHK21 cells)	ND	[85]
	Quercetin	Virucidal activity; inhibits adsorption of the virus	~212.1 µg/mL; ~5.8 µg/mL (Vero cells) ^$^	ND	[118]
	Biacalein				
	PI 88	Creates steric hindrance to the JEV attachment; immunomodulatory action	40 µg/mL (BHK21 cells)	~40% (C57B1/6)	[85]
	Methyl-β-cyclodextrin	Inhibits viral replication and entry in the host due to the depletion of cholesterol	ND *	ND	[119]
	Filipin III				
**Viral protein inhibitors**	Furanonapthoquinone	Inhibits viral RNA and protein synthesis/expression	ND *	ND	[120]
	Amphoterecin B	Inhibits the viral replication and protein synthesis/expression	7.8 µg/mL (BHK21 cells)	ND	[121]
	Suramin	Blocks production of the viral E and NS3 proteins	50 µg/mL (BHK21 cells)	ND	[85]
	Niclosamide	Inhibitsthe NS2B-NS3 protease; endosomal acidification	5.80 µM (BHK21 cells)	ND	[122]
	SK-12 protein	NS2B-NS3 inhibitors	~29.81 µM (Vero cells)	ND	[123]
	ARDP0006				
	Temoporfin		0.011 µM (HDF9 and hNPCs cells)	ND	[59,124]
	NSC 12155	NS5 inhibitor	1.4 µM (BHK21 cells)	ND	[125]
	N-methylisatin-beta-thiosemicarbazone derivative (SCH 16)	Inhibits early translation	16 µg/mL (PS cells)	100% (Swiss albino)	[126]
	Scopolamine hydrobromide	Binds to the active site of NS5, thus inhibiting the JEV replication	ND *	ND	[127]
	N-nonyl-deoxynojirimycin (*N*N-DNJ)	Inhibits α-glucosidase enzymes causing misfolding of viral proteins	ND *	~54% (ICR mice)	[60]
	Belladonna	Reduces the NS3 protein caspase 3 and 8 enzymatic activity and its expression	7.01 µg/mL (CHME3 and SHSY-5Y cells)	ND	[128]
	Pentoxyfylline	Interferes with the assembly and release of virions	50.3 µg/mL (PS cells)	100% (Swiss albino)	[74,129]
	Manidipine	Inhibits NS3 Helicase, targets NS4B and calcium channel	1.6 µM (Vero cells)	80% (BALB/c)	[28]
	Nitazoxanide	Activates elF2α; targets the JEV replication at the early mid stage	0.12 µg/mL (BHK21 cells)	~70–90% (Chinese chumming mice)	[130,131]
	Luteolin	Inhibits synthesis of the E protein	4.56 µg/mL (A549 cells)	ND	[132]
	Erythrosine B	Inhibits flaviviral NS2B-NS3 protease	0.35 µM (A549 cells)	ND	[133]
	Ivermectin	Inhibits the NS3 helicase	0.3 µM (Vero cells)	ND	[134]
	Andrographolide	Inhibits the NS3 protease	IC50 = 2 µg/mL (Enzymatic assay)		[135]
	Mycophenlate, P5	Inhibits the E protein	3.1 µg/mL (PS cells)	~75% (Swiss albino)	[136,137]
	Monoclonal antibodies (1H7, 2D4, 3C4, 3H7, 3F10)	NS3 and NS5 inhibitors	ND *	ND	[138]
	NITD008	NS5 polymerase inhibitor	3.09 µM (BHK-21 cells)	ND	[139]
**Viral replication inhibitors**	Lonafarnib	Inhibits viral replication	0.982 µM (Huh-7 cells)	ND	[139]
	Nitroxoline		2.482 µM (Huh-7 cells)	ND	
	Cetylpyridinium chloride		0.35 µM (A549 cells)	ND	
	Cetrimonium bromide		2.232 µM (Huh-7 cells)	ND	
	Hexachlorophene		0.421 µM (Huh-7 cells)	ND	
	Cilindipine		3.5 µM (Vero cells)	ND	[28]
	FGIN-1-27		3.21 µM (BHK21 cells)	ND	[140]
	Ribavirin	Inhibits synthesis of gunanine nucleotides	3.9 µg/mL (PS cells)	ND	[72]
	2-Deoxy-D-glucose and 3-deazauridine	Interferes in the synthesis of RNA, DNA and glycoprotein of JEV	ND *	ND	[141]
	MCPIP1 (Monocyte chemoattractant protein 1-induced protein 1)	Targets various RNA sites and inhibits replication	ND *	ND	[87]
	Pokeweed protein (Phytolacca americana)	Depurinates viral RNAs	300 ng/mL (BHK-21 cells)	~85% (BALB/c)	[89]
	Diadzin	Binds to the frameshift site in RNA; inactivates the virus	25.9 µM; 40.4 µM (BHK21 cells) ^$^	ND	[90]
	Kaempferol				
**Virus assembly and maturation inhibitors**	10,10′-bis (trifluoromethyl) marinopyrrole A thiophene	Inhibits the proliferation of JEV	0.05µM (BHK21 and RD cells)	ND	[142]
	Nelfinavir	Protease inhibitor	1.6 µM (Vero cells)	ND	[28]
	Palmatine	Protease inhibitor	ND *	ND	[143]

ND: in vitro and/or in vivo efficacy was not determined. ND *: in vitro efficacy determined by performing a viral inhibition assay but IC_50_ not evaluated. DM ^#^: Delayed mortality observed but % survival was not estimated. ^$^ in vitro efficacy were determined in two different assay types (pre and post treatment of JEV infection).

## 9. JEV Vaccine

Due to the dearth of therapeutics, vaccination seems to be a reliable means of prevention, other than avoiding mosquito bites [144]. As part of pre-travel precaution in most countries, multiple doses of JEV vaccines are recommended [145]. Since the 1990s, significant progress has been made in JE surveillance and the implementation of vaccination programs. In 2012, 75% of countries with a JE transmission risk had surveillance programs in place, and 46% had immunization programs [146]. Large-scale vaccination of the susceptible human population must be implemented in order to prevent JE. There are several types of JE vaccines, including purified, formalin-inactivated mouse-brain derived, cell-culture derived inactivated and cell-culture derived live attenuated vaccines [9] (Table 2).

### 9.1. JE-MB

The first-generation inactivated Nakayama strain JE-MB [marketed as ‘JE-VAX’ or ‘Biken’] vaccine derived from mouse brain was manufactured in Japan and was licensed in 1954 [9]. In Taiwan, the first trial of the crude vaccine was conducted in 1965, where an 80% effectiveness was observed. Later, adults from India, Japan, Thailand and the US reported 80% to 100% sero-protection after receiving a purified vaccine. The Beijing-1 strain, also known as P1, replaced the Nakayama strain in 1988 after it exhibited better cross-neutralization and broader coverage on different JEV strains [147]. Based on the JEV Beijing-3 (P3) strain, in 1967, a primary hamster kidney [PHK] cell culture-derived inactivated vaccine was produced in China with fewer adverse effects [9]. The vaccination was recommended by the Centers for Disease Control and Prevention (CDC) for the age group 1–3 years, in a three-dose regimen at 0, 7 and 30 days which displayed better neutralizing antibodies in 100% of the recipients within 6 months. Immunity against JEV persisted in some recipients even without the booster for two years, although the booster is recommended [9,148]. Later in 2005, the Japanese government retracted this vaccine from their immunization plan when it was temporarily linked to the risk of acute disseminated encephalomyelitis (ADEM) along with other hypersensitive responses, that occurred at the rate of 1–17/10,000 vaccinated people. The WHO Global Advisory Committee on Vaccine Safety (GACVS) subsequently clarified and explained that the JE-MB vaccine had no causal relationship with an elevated risk for ADEM [149]. The Nakayama strain of JEV had been the predominant strain utilized in JE-MB production throughout Asia since it was identified from a patient’s CSF in 1935 and was maintained via continual mouse brain passage [150]. However, cell-culture-based vaccinations have mostly supplanted the JE-MB vaccine [151,152].

### 9.2. JE-VC

JE-VC is a Vero cell culture-derived vaccine made up of an inactivated SA 14-14-2 strain of JEV with 0.1% of aluminum hydroxide as an adjuvant [148]. In the United States and most other countries the JE-VC is marketed as IXIARO^®^. However, in Australia and New Zealand, it is sold as JESPECT^®^ and in India its brand name is JEEV^®^ [153]. The JE-VC was initially manufactured by Intercell biomedical, Austria and licensed in 2009 for aged 17 years and above. Later in 2013, it was further licensed for age ≥2 months to 16 years [154] and reported 100% seroconversion in vaccinated individuals with the booster dose. However, some cases also reported a decline in the seroconversion rate with time [155,156]. This vaccine has the advantage of a higher immunogenicity and greater antibody titer, as compared to the mouse-brain derived JE-MB vaccine. The most common side effects of this vaccine were identified as headache, myalgia, influenza-like sickness and fatigue in clinical trials in 13–26% of participants in the first week of immunization [148]. The other variant of JE-VC vaccine, which is made up of the inactivated Beijing-1 strain of JEV without adjuvant, was licensed in Japan as JEBIK-V and ENCEVAC in 2009 and 2011, respectively. TC-JEV also has the inactivated Beijing-1 strain that was manufactured in Boryung/Korea and licensed in 2013 [153].

### 9.3. JE-CV

IMOJEV is a recombinant live chimeric JE vaccine (JE-CV), based on the 17D-204 yellow fever vaccine, where two structural genes (prM and E) were replaced with the JEV prM and E genes of an SA 14-14-2 attenuated strain [157]. It was designed by Chamber’s group in 1999, and further developed by Guirakhoo’s group (Acambis, Cambridge, MA, USA) as a ChimeriVax™-JE vaccine. Thereafter, it was licensed in Europe, USA and Australia with the brand name IMOJEV in 2009, which is manufactured by Sanofi Pasteur [158]. This single-dose live vaccine induces a significant immune response with ~95% seroconversion rate [9] and is recommended for immunization in age group ≥9 months and ≤18 years followed with a booster dose after 1–2 years of priming. Later on, the vaccine was recommended for adults (>18 years) with a single dose followed with an optional booster dose after 5 years in JEV endemic areas [147]. Headache, weariness, myalgia and malaise were the most common side effects observed in its clinical trials in 17–24% of vaccine recipients. To date, the JE-CV vaccine is approved in 14 countries including Thailand, Australia, Malaysia, Philippines, Hong Kong and Singapore [159].

### 9.4. JE-LV

Live attenuated Japanese encephalitis vaccine (JE-LV) is made up of an attenuated strain of SA 14-14-2 virus and produced from the primary culture of hamster kidney cells [160]. The vaccine was initially developed by CDIBP [Chengdu Institute of Biological Product], China in 1988. This is a single-dose live vaccine, recommended for children above the age of 8–9 months with a booster dose after 3–12 months. The vaccine has exhibited an 85–95% efficacy with single, as well as double doses in reported trials [147]. According to the WHO, neutralizing antibodies are produced in ~90% of vaccine recipients [147] and confer protection against JE infection at least for 5 years. Approximately more than 700 million doses of the JE-LV vaccine have been given out globally since 1988. Although it is an effective vaccine, it has displayed various side effects, including fever, drowsiness, irritability, nasopharyngitis, gastroenteritis, conjunctivitis and rhinitis in the clinical trials [161]. JE-LV has the largest global production share of almost 50% among all JE vaccines, including its presence in West Pacific regions and a few Asian countries [Nepal, Sri Lanka, India and Korea] [161]. Owing to its potential, it has been used more frequently in JE endemic areas since 2003 [29].

### 9.5. JENVAC

JENVAC is a Vero cell adapted inactivated vaccine developed from the Indian Kolar strain (821564XY) which is manufactured by the Indian pharma company, Bharat Biotech International Ltd., Hyderabad, India, and was licensed in 2014 [162]. This vaccine provided >90–96% seroconversion and seroprotection after 28 days of immunization in clinical trials, which were conducted in healthy individuals aged 1–50 years with the minimal recorded adverse effects, such as, fever, headache, vomiting, pain at the injection site and body ache [162]. JENVAC is administered in two-doses with 24 months interval and exhibits the highest antibody titer. It has proven a better immunogenicity over the live attenuated vaccine made with (SA-14-14-2). It is recommended for adults, as well as children of age groups ≥1 year [163].

## 10. Discussion

Despite numerous studies that have improved our comprehension of the virus and how it interacts with the host, JE still poses a serious threat to public health and has the potential to spread globally. Coordinated efforts are required for JE control and management, ranging from mosquito control to the design of specific antiviral drugs and effective vaccines. The majority of JEV outbreaks have taken place in developing countries. Therefore, finding drugs that could reach the underprivileged masses is the obligation of the scientific community, federal governments and the WHO. Drugs developed must be safe for all, including children, the elderly and immuno-compromised people. These drugs must cross the blood–brain barrier (BBB), reach the central nervous system and remain active after the initial stage of the infection, even when the patient begins to show the symptoms of the disease, and must possess a high genetic barrier to resistance.

Novel antiviral drug development and clinical application must be a significant area of focus in the current scenario. Antivirals with fewer off-target effects include those that inhibit the activity of conserved non-structural viral proteins, such as protease and polymerase. The availability of JEV NS3 and NS5 crystal structures has made it possible to identify potential inhibitor-binding sites for developing the most effective drugs using structure-guided target-based drug development [140,164,165,166]. It may be possible to develop antivirals from specific inhibitors targeting host proteins necessary for any stage of the viral life cycle, from attachment and entry to replication, maturation and its release. An active area of research that could produce safe and effective antiviral drugs involves high throughput screening of drug libraries and natural compounds [139,167]. Furthermore, research focused on dengue virus (DENV), zika virus (ZIKV) and other similar viruses could be extrapolated to design an effective JEV-specific drug. Studies have shown that several types of phytochemicals, including flavonoids, terpenoids, polysaccharides, alkaloids, thiophenes, lignans and lectins, have significant antiviral activities. Extensive research is required to fully explore the potential for natural products to be developed as anti-JE drugs. In addition to having a synergistic antiviral effect, combining different drugs/compounds is an intriguing way to create optimal anti-JE therapies. This strategy allows for a lower dose of each antiviral treatment while maintaining its efficiency and, thus, a lower level of cytotoxicity. Different combinations of multiple drugs from various pharmacological categories need to be evaluated in in vitro followed by in vivo studies to discover effective formulations.

A JEV-specific monoclonal antibody is one example of a prospective treatment. If effective, it might serve as a clinical proof-of-concept for developing treatments for other arboviral encephalitis. This approach has the potential to hasten the development of therapies for emerging flaviviruses. It is anticipated that novel vaccines with excellent safety profiles and improved immunogenicity will be available in the near future which may restrict JEV infection in new areas. Some examples include the experimental Japanese encephalitis vaccines that have already reached clinical studies and the next-generation vaccines that could be produced by utilizing adaptable vaccine platforms, similar to the strategy recently used for the development of a COVID-19 vaccine. With highly advanced biomedical science technologies at our disposal, the future holds out hope for establishing effective JE treatments and immunization with high coverage.

## Figures and Tables

**Figure 1 vaccines-11-00742-f001:**
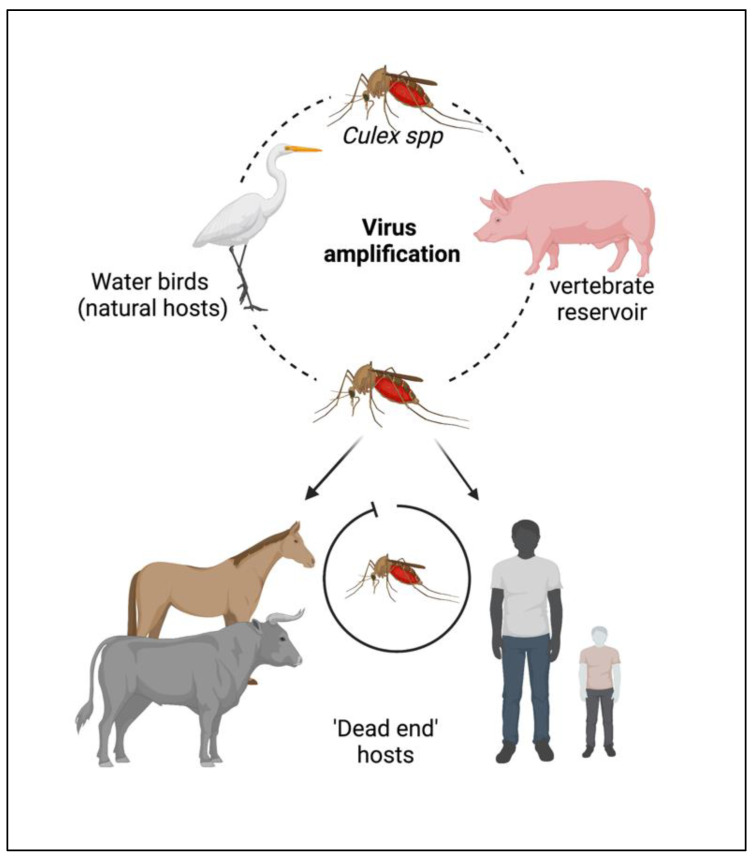
Cycle of the Japanese encephalitis virus (JEV) infection and amplification. Long-legged water birds, such as herons, storks, and ibises are the primary reservoirs and natural hosts for JEV. JEV-infected female mosquitoes, especially Culex tritaeniorhynchus, transmit the virus from wading water birds to other animals (pigs, cattle, and other hooved animals) and humans. Pigs act as secondary hosts where the virus becomes amplified at an optimum level and carries the infectious virion from one place to another (vector-free transmission). Female Culex mosquitos take up the virus from here and infect humans by biting them. The infected humans act as ‘dead-end’ hosts for the virus as JEV does not develop a titer high enough in the blood circulation to transmit through feeding mosquitoes.

**Figure 2 vaccines-11-00742-f002:**
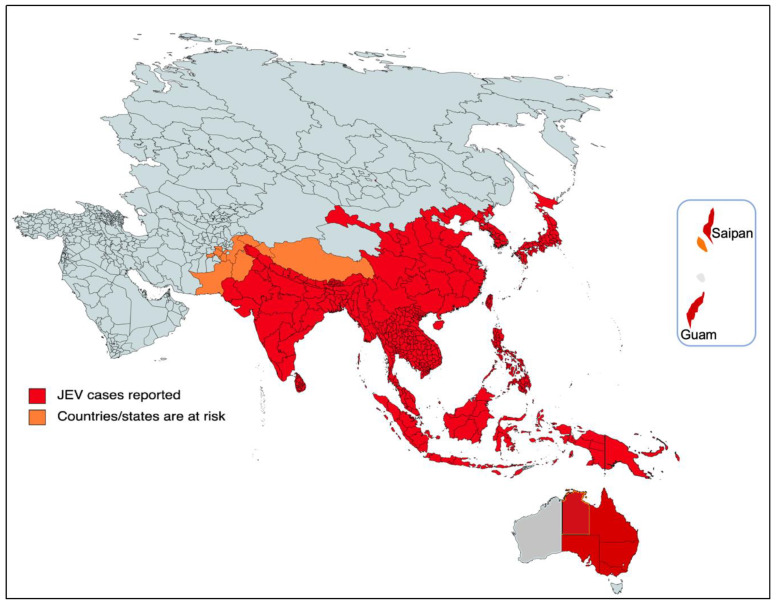
Geographical distribution of JEV. The red part of the focused Indo-Pacific geographical regions and countries indicates where active JEV cases have been reported since its outbreak. The orange color shows the areas that have the highest risk of JEV infection in the near future. The epidemiological data of JEV was modified from https://www.who.int/news-room/fact-sheets/detail/japanese-encephalitis (accessed on 26 December 2022).

**Figure 3 vaccines-11-00742-f003:**
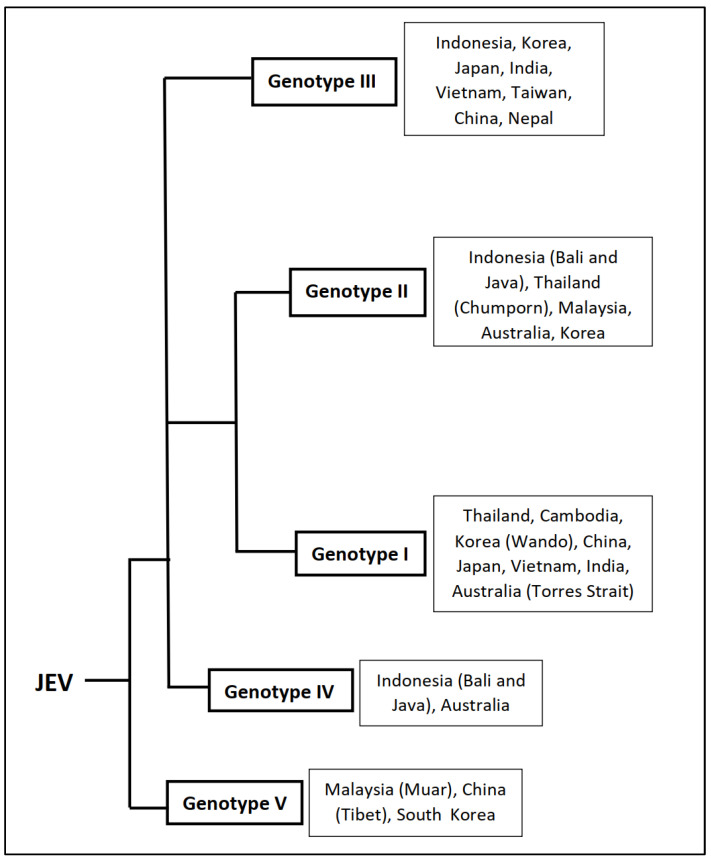
Genotypes of JEV and most affected countries. Based on the nucleotide sequences of the C/prM and E protein genes, JEV is classified into five genotypes as Genotype I, II, III, IV and V. A general representation of the origin of the various JEV genotypes, along with the countries/geographical areas where they are most abundant in or have the greatest impact (in the right side of the respective genotypes). The data represented is adapted from van den Hurk et al., 2022 [18], Gao X et. al., 2015 [31], Solomon T et. al., 2003 [32].

**Figure 4 vaccines-11-00742-f004:**
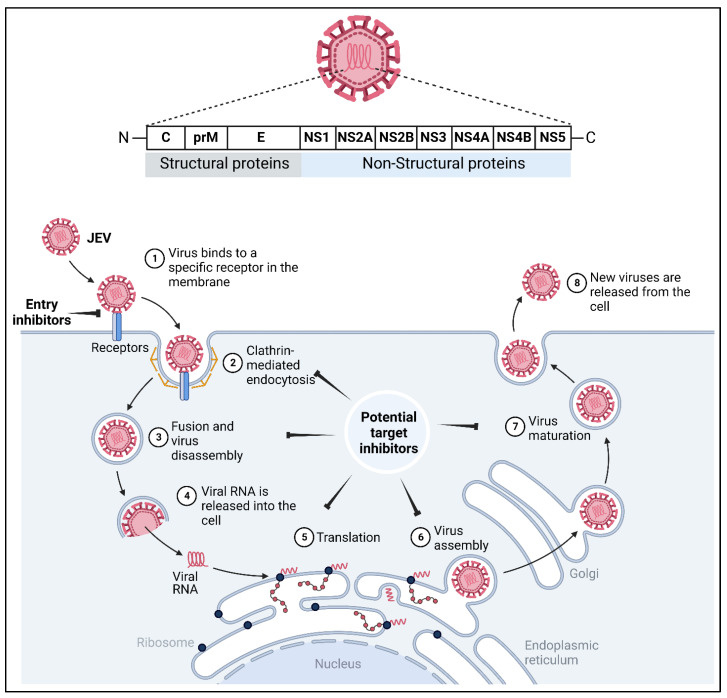
JEV structure, pathogenesis and potential drug targets. Schematic representation of JEV, comprising three structural proteins, namely capsid (C), precursor membrane (prM) and envelope (E), and seven non-structural (NS) proteins, NS1, NS2A, NS2B, NS3, NS4A, NS4B and NS5. N and C represent amino- and carboxyl-termini of the viral polyprotein. The pathogenesis of JEV begins with the binding of the virion onto the host receptor (DC-SIGN, heparin sulphate, Fc-receptor, etc.) and the internalization through clathrin-mediated endocytosis. Thereafter, the fusion of the viral particle/virion with endosome brings about a conformational change which releases the genetic material in an acidic environment. Very soon, the translation of viral RNA begins with the aid of host machinery onto a rough endoplasmic reticulum (ER) resulting in the synthesis of structural and non-structural proteins. In cognate, the NS viral proteins help in the formation of the replication complex and the genomic positive sense RNA becomes transcribed into complementary negative sense RNA. Furthermore, NS proteins (majorly NS3, NS4B and NS5) form a replication complex, and further several copies of positive sense genomic RNA become transcribed which are later encapsulated by the capsid protein, assembled with prM and E onto ER, and the assembled virus matures over the trans-Golgi network and finally the matured virions are released from the host cell through exocytosis. The inhibitory symbols indicate the potential drug targets for which efforts in the development of antivirals have been made so far, and research to diminish the JEV infection continues. Figure was originally created from BioRender.com (accessed on 26 December 2022).

**Table 2 vaccines-11-00742-t002:** Licensed JEV vaccines so far.

Licensed Vaccines (Types)	Utilized Viral Strain	Trade Name	Licensing Year	Required Doses	Vaccination Age	Route of Administration	Countries Licensed to Use
**JE-MB ^⌂^** **(Inactivated mouse brain-derived JE vaccine)**	Nakayama-NH Beijing-1	JE-VAX	1954 1993	3	≥12 months	Subcutaneous	European Union, India, Japan, Malaysia, North Korea, South Korea, Sri Lanka, Taiwan, Thailand, United States, Vietnam
**JE-VC (Inactivated Vero cell culture-derived JE vaccine)**	SA14-14-2 (with adjuvant)	IXIARO^®^/ JESPECT^®^/ JEEV^®^	2009 * 2013 *	2	≥17 year ≥2 month	Intramuscular	Australia, Bangladesh, Bhutan, Canada, European Union, Hong Kong, India, Japan, Latin America, Nepal, New Zealand, Pacific Islands, Papua New Guinea, Singapore, South Korea, Switzerland, United States
	Bejing-I (without adjuvant)	JEBIK-V ENCEVAC TC-JEV	2009 2011 2013	3	≥36 month	Subcutaneous	Japan, South Korea
**IMOJEV (Recombinant chimeric virus vaccine)**	SA14-14-2	IMOJEV	2010	1	≥1 year	Subcutaneous	Australia, South Korea, Thailand
**JE-LV ^⌂^ (Live attenuated Japanese encephalitis vaccine)**	SA14-14-2	CD.JEVAX	1988	1	≥8 months	Subcutaneous	China, Hong Kong, India, Japan, Nepal, South Korea, Sri Lanka, Thailand
**JENVAC**	821564XY (Indian Kolar strain)	JENVAC	2013	2	≥6 months	Intramuscular	India

^⌂^ JEV vaccines discontinued. * Licensed in 2009 for aged ≥ 17 year, later on it was approved for ≥2 month-old children in 2013.

## Data Availability

Not applicable.

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
