# Peer review of "Japanese Encephalitis Virus: An Update on the Potential Antivirals and Vaccines"

_vaccines, 2023, doi:10.3390/vaccines11040742_

Round 1

Reviewer 1 Report

The review covers various aspects of JEV, However I have the following comments/suggestions:

The title of the review does not go with the content in the review. From the title it is expected that the authors would be stressing more on the situation of JEV in the Indo-Pacific Region but that does not seem to be the case. I suggest authors to consider another title for the review.

The message from lines 53-54 is not clear.

As authors mention in Figure 2 " orange color is showing areas which have highest risk of JEV infection in near future" a description on how this prediction was made and on what basis would be informative.

At various instances, authors have mentioned "Indian continent". This needs to be corrected.

Figure 3. What do authors mean by "Spherical representative diagram"?

Figure 3 needs to be improved. It can be represented in a more descriptive way.

The authors need to work on figure legends. A lot of grammatical mistakes are observed.

The table legends should be improved.

The authors should try to include latest data and statistics in the review.

Author Response

We thank the Reviewer for reviewing our manuscript critically which improves the overall manuscript. The point-by-point response/justification to the reviewer is attached herewith. 

Reviewer 2 Report

Japanese Encephalitis Virus: An Emerging Threat in the Indo-Pacific Region

It is a systematic review which highlights the current status of efforts put in to develop antivirals against JE and the available vaccines along with their effectiveness. It summarizes epidemiology, structure, pathogenesis, and potential drug targets that can be explored to develop a new range of anti-JEV drugs to combat JEV infection globally.

It is a nice and useful review.

Minor mistakes below indicated can be managed and improve the quality of the paper.

I recommend minor revision

REVIEW

le Culex spp. mosquito. JE shows l.12

pl spp not in cursive

An. Peditaeniatus and An. hyrca- l.47

An. Pedxxxx

L 32, 48 genus and family not in cursive

References

Pl check one by one and revise (edit). There are many irregularities. Pl adapt to the journal’s guideline

Author Response

We thank to the reviewer for reviewing our manuscript which improved the
overall manuscript. The point-by-point justification/response is as follows:

Q. 1: English language and style are fine/minor spell check required.
Ans.: Thanks for raising the English language and spelling check concern. We
have thoroughly edited for proper English language, grammar, punctuation,
spelling and overall style in the revised manuscript.
Q. 2: Minor revision…
Ans.: All the minor revisions were well taken in the revised manuscript. All the genus and species names are properly formatted.
Q. 3: References…
Ans.: All the references are arranged according to the Journal guidelines by using “Mendeley reference manager”.

Reviewer 3 Report

This review on japanese Encephalitis virus by Kumar Saurabh Srivastava and colleagues is well written and complete. It describes well the virus, its biology, its epidemiology as well as the therapeutic strategies and vaccine prevention that can be used.

I have only a few comments :

-          The article still needs to be carefully proofread for english editing and typos.

-          The alternation between JEV and JE is confusing. JEV could be the abbreviation chosen throughout the paper to be homogeneous.

-          Some information is given twice such as the nature of the virus genome.

-          The part describing viral replication could be removed from the chapter on pathogenesis and could constitute an independent chapter following the description of the virus and its genome.

-          The section on clinical manifestations should also describe the biological abnormalities observed during Japanese encephalitis in the blood as well as in the CSF.

-          The chapter on genetic diversity would be better placed between the description of the virus and its genome, and the newly created chapter on viral replication. The authors should discuss, in this chapter, the respective virulence of the different genotypes and the molecular determinants of this virulence if they are known.

-          The paragraph on potential drug targets does not seem necessary to me and only repeats elements of the description of the function of the various viral proteins described in the structure and genome chapter. The information contained in chapter 7 could therefore be repatriated at the end of chapter 3.

Author Response

We thank the reviewer for reviewing our manuscript which improved the
overall manuscript. The point-by-point justification/response to the reviewer is attached herewith. 

Round 2

Reviewer 1 Report

The authors response is majorly convincing however the text needs to be improvised further with respect to English grammar and style. For example: Japanese Encephalitis (JE) was reported for the first time in 1871 from Japan and hence the name. This statement (Lines 36-37) needs to be reframed as the information looks incomplete.

 Pigs act as..Figure legend 1

 etc.

As the authors have changed the title of the review, a detail of JEV outbreaks from India is not required. I would suggest authors should make that precise.

As the title of the review says ......an update on the potential antivirals and vaccines, if possible, authors should discuss a little about the available antivirals and vaccines at the beginning of the review which can be detailed later. This would help the readers to connect with the title of the review.

Author Response

We thank to the reviewer for reviewing our manuscript and suggested for few more minor edits which improved the overall manuscript. The point-by-point response against the raised concerns are as follows:

Q. 1: Text needs to be improvised further with respect to English grammar and style.

Ans.: Thanks for raising the concern of English language and spelling check. We have thoroughly edited the manuscript again for proper English language, grammar, punctuation, spelling and overall style in the revised manuscript.

Q. 2: Pigs act as…..Figure legend 1.

Ans.: Thank you for your suggestion to edit figure legend 1. We have edited the figure legend as per the suggestion.

Q.3: As the authors have changed the title of the review, a detail of JEV outbreaks from India is not required. I would suggest authors should make that precise.

Ans.: As per the suggestion, we have thoroughly edited the whole manuscript wherever was required. Also, we have concise the epidemiology section particularly related to the Indian perspective (lines 162-170).

Q.4: As the title of the review says ......an update on the potential antivirals and vaccines, if possible, authors should discuss a little about the available antivirals and vaccines at the beginning.

Ans.: We thankful to the reviewer for this issue. In the revised manuscript, we have discussed about the antivirals and vaccines precisely at the beginning (lines 112-120).